**∂ | Open Peer Review** | Physiology and Metabolism | Research Article

# DegQ is an important policing link between quorum sensing and regulated adaptative traits in *Bacillus subtilis*

Tjaša Danevčič,[1] Mihael Spacapan,[1] Anna Dragoš,[1] Ákos T. Kovács,[2] Ines Mandic-Mulec[1]

**ABSTRACT**   Quorum sensing (QS) is a widespread bacterial communication system that controls important adaptive traits in a cell density-dependent manner. However, mechanisms by which QS-regulated traits are linked within the cell and mechanisms by which these links affect adaptation are not well understood. In this study, *Bacillus subtilis* was used as a model bacterium to investigate the link between the ComQXPA QS system, DegQ, surfactin and protease production in planktonic and biofilm cultures. The work tests two alternative hypotheses predicting that hypersensitivity of the QS signal-deficient mutant (*comQ::kan*) to exogenously added ComX, resulting in increased surfactin production, is linked to an additional genetic locus, or alternatively, to overexpression of the ComX receptor ComP. Results are in agreement with the first hypothesis and show that the $P_{srfAA}$ hypersensitivity of the *comQ::kan* mutant is linked to a 168 strain-specific mutation in the $P_{degQ}$ region. Hence, the markerless Δ*comQ* mutant lacking this mutation is not overresponsive to ComX. Such hyper-responsiveness is specific for the $P_{srfAA}$ and not detected in another ComX-regulated promoter, the $P_{aprE}$, which is under the positive control by DegQ. Our results suggest that DegQ by exerting differential effect on $P_{srfAA}$ and $P_{aprE}$ acts as a policing mechanism and the intracellular link, which guards the cell from an overinvestment into surfactin production.

**IMPORTANCE**   DegQ levels are known to regulate surfactin synthesis and extracellular protease production, and DegQ is under the control of the ComX-dependent QS. DegQ also serves as an important policing link between these QS-regulated processes, preventing overinvestment in these costly processes. This work highlights the importance of DegQ, which acts as the intracellular link between ComX production and the response by regulating extracellular degradative enzyme synthesis and surfactin production.

**KEYWORDS**   quorum sensing, surfactin, DegQ, extracellular protease, biofilm, *Bacillus subtilis*

I n a process called quorum sensing (QS), bacteria communicate using small, diffusible molecules. Above a threshold concentration, these molecules bind to cognate receptors and induce coordinated changes in gene expression within the population. Many QS systems have been studied in details at the molecular level (1–3), but recently they are also being applied as models to address bacterial social behaviors in populations represented by mixtures of QS mutants and wild-type strains (4–8). QS-controlled processes may rely on additional regulatory links, embedded within QS networks that safeguard investment in metabolically expensive processes, such as QS (4), but these links are only partially understood. In the Gram-positive bacterium *Bacillus subtilis*, the *comQXPA* locus encodes the major peptide-based QS system (9). The activity of this system is modulated by a QS-signaling peptide ComX (5, 10–15). ComX is modified and activated by the isoprenyl transferase ComQ (16–18), and its extracellular accumulation

Address correspondence to Ines Mandic-Mulec, Ines.MandicMulec@bf.uni-lj.si.

The authors declare no conflict of interest.

See the funding table on p. 12.

leads to the phosphorylation of ComP and consequently induction of the ComA regulon (19, 20). Phosphorylated ComA (ComA-P) directly activates the *srfA* operon, responsible for surfactin synthesis, and an embedded small gene coding for the ComS protein, which ultimately prevents ComK degradation and thus pushes cells into the K-state or genetic competence for transformation (21, 22).

Our previous work demonstrated that the QS signal-deficient strain of *B. subtilis* (*comQ::kan*) has a decreased surfactin production, and it grows faster than the wild-type strain (5). Moreover, when this mutant was supplemented with the purified QS signal ComX or co-cultured with a signal producer, it overproduced surfactin and its population fitness drastically decreased compared with the mutant monoculture (5). However, the mechanisms leading to the surfactin overproduction phenotype by the ComX-supplemented signal-deficient mutant are not completely understood (5, 7). We tested here two hypotheses. First, we asked whether the decreased fitness of the *comQ::kan* mutant exposed to ComX is linked to an unidentified genetic locus, which is responsible for surfactin overexpression and partial cell lysis, and thus serves as the link between QS sensing and the response (5). Second, we tested whether hypersensitivity of the *comQ::kan* mutant to ComX arises from a polar effect of the inserted kanamycin resistance gene (*kan*) into the *comQ* gene, thus increasing expression of the downstream *comP* receptor gene. The increased abundance of ComP receptors would then amplify the QS response, which is costly (7).

To address the two hypotheses, we initially investigated whether the hypersensitivity of the *comQ::kan* mutant to ComX is mimicked in the markerless Δ*comQ* mutant at the level of *srfA* expression. We next compared the Δ*degQ* and P$_{degQ168}$ mutant strains. In *B. subtilis*, the ComQXPA QS system also activates transcription of the *degQ* gene, which activates genes for extracellular degradative enzymes (e.g., *aprE*) via the two-component regulatory system, DegU/DegS (19, 23–27). As DegQ mediates de-repression of the *srfA* operon (28, 29), we hypothesized that it may be involved in the hypersensitivity of the *comQ::kan* mutant to ComX, which has not been directly addressed yet. Moreover, we evaluated the polar effect of the kanamycin resistance gene (*kan*) insertion into the *comQ* gene, which may render the strain hypersensitive to ComX by increasing the transcription levels of the *comP* gene. Our work affirms that the *degQ* locus, while unlikely the increased *comP* transcription, leads to hypersensitivity of *comQ::kan* mutant and suggests that DegQ serves as a link between QS and the response.

## RESULTS

### Sensitivity to ComX signal differs between distinctly constructed signal-negative mutant *comQ::kan* vs markerless *comQ* mutant

*B. subtilis* PS-216 is a natural isolate amenable to genetic manipulation due to its ability to develop natural competence and is often used to investigate biofilm formation and microbial interactions (5, 15, 26, 30). The signal-deficient *B. subtilis* PS-216 mutant (*comQ::kan*) was previously reported to be hypersensitive to the ComX signal (5). To determine the reason for the hypersensitivity of this strain, we compared the phenotypes of the PS-216 *comQ::kan* mutant (5, 7, 16) and the PS-216 Δ*comQ* mutant, which carries a complete deletion of the *comQ* gene without an antibiotic resistance marker (26). Neither strain secretes an active signaling peptide ComX, as previously demonstrated (16, 26, 30). In addition, we also included PS-216 Δ*degQ* and PS-216 P$_{degQ168}$ strains in the comparative analyses since ComQXPA QS system also activates transcription of the *degQ* gene (19, 23–26). Notably, lack of DegQ contributes to the de-repression of the *srfA* operon (28, 29). However, to the best of our knowledge, the role of DegQ as the link between ComX-sensing and the QS response has not been directly addressed previously.

We initially compared the expression from P$_{srfAA}$ in different *B. subtilis* PS-216 strains (wt, P$_{degQ168}$, Δ*comQ*, *comQ::kan*, and Δ*degQ*) (Fig. 1). Expression from P$_{srfAA}$ in the Δ*comQ* mutant without exogenous ComX was low but increased significantly (*P* = 0.02) in the presence of exogenous ComX, reaching the levels produced by the wild-type strain. In contrast, expression from P$_{srfAA}$ in the *comQ::kan* mutant was similar to that of

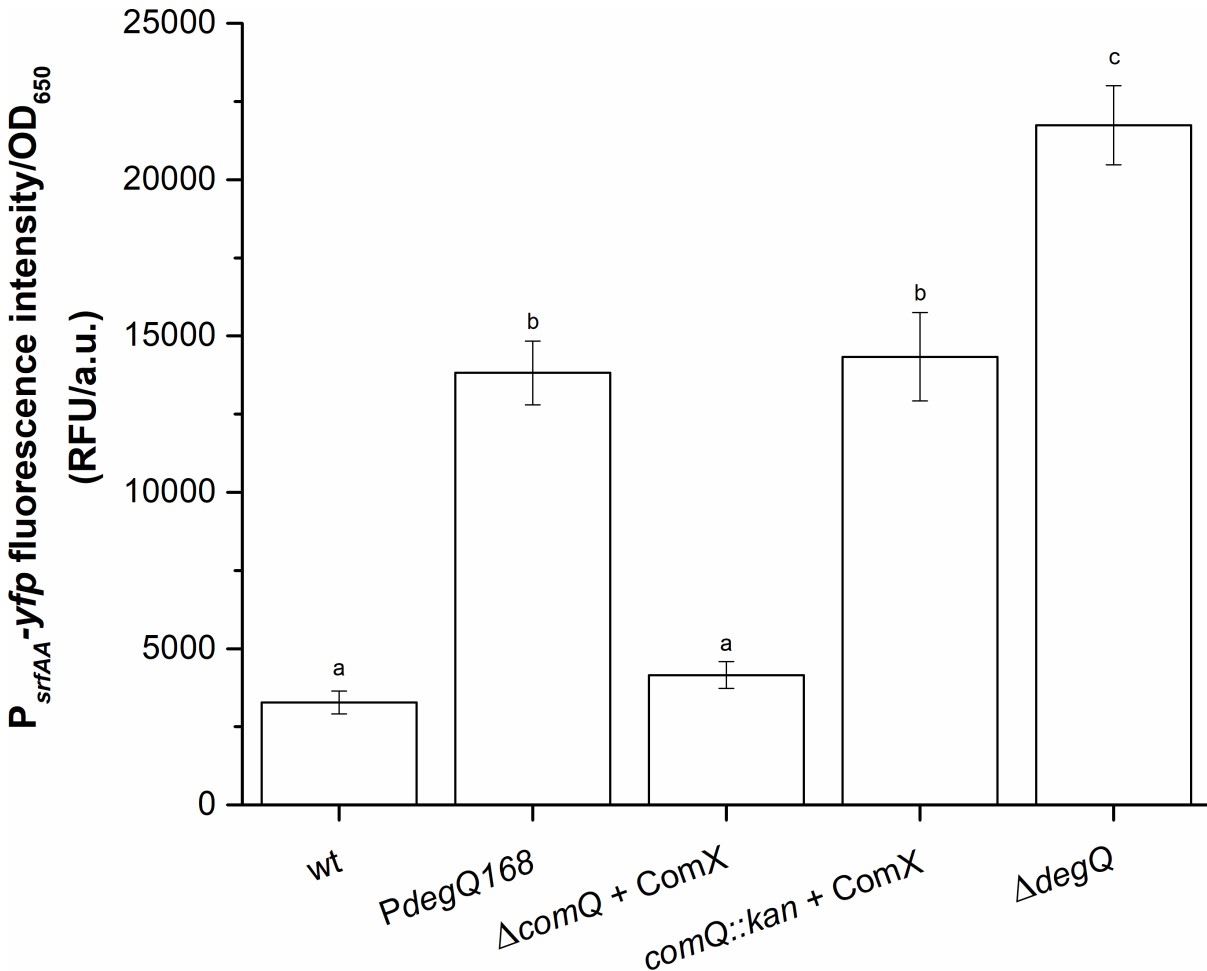

**FIG 1** Expression (fluorescence intensity normalized per optical density at 650 nm) derived from the $P_{srfAA}$-*yfp* constructs of *Bacillus subtilis* PS-216 wild-type strain and its mutants (mutant with point mutation in $P_{degQ}$: $P_{degQ168}$; QS signal-deficient mutants: Δ*comQ* and *comQ::kan*; extracellular protease production-deficient mutant: Δ*degQ*) after incubation in CM medium at 37°C and 200 rpm for 6 h. Mutants Δ*comQ* and *comQ::kan* were grown in CM medium supplemented with ComX-conditioned 5% (vol/vol) M9 minimal medium, where ComX is produced heterologously in *E. coli* ED367 after IPTG induction. The values presented are means and standard errors ($n = 3$). Different letters above the columns indicate a statistically significant difference ($P < 0.05$) between mean values across all strains.

the wild-type strain without exogenous ComX ($P = 0.16$) but further increased (three-to fourfold) in the presence of exogenous ComX. Therefore, the response of the Δ*comQ* mutant to ComX did not mimic that of the *comQ::kan* strain. Moreover, a significantly increased expression was detected from $P_{srfAA}$ in the Δ*degQ* mutant (Fig. 1), which was approximately 1.5-fold higher compared with the *comQ::kan* mutant supplemented with exogenous ComX. This phenotype resembled that of *B. subtilis* 168 strain, bearing a point mutation in $P_{degQ}$ ($P_{degQ168}$) (Fig. S1), which produces less DegQ and leads to overly active $P_{srfAA}$ and higher competence for transformation than *B. subtilis* NCIB 3610, as previously reported in reference (28).

**Increased sensitivity of signal-negative mutant *comQ::kan* is linked to point mutation in *degQ* promoter, horizontally transferred from domesticated *B. subtilis* strain**

To further confirm our hypothesis that the point mutation in $P_{degQ}$ is responsible for the ComX-dependent hypersensitive and hyper-competent phenotypes of the *comQ::kan* mutant, we swapped $P_{degQ}$ of the wild-type *B. subtilis* PS-216 strain with that of *B. subtilis* 168 (Fig. 1). Expression from the $P_{srfAA}$ was approximately 4.5-fold greater in strains

with $P_{degQ168}$ (Fig. 1) compared with the wild-type PS-216 strain. Moreover, $P_{degQ}$ was sequenced in the experimental strains (wt, ΔcomQ, comQ::kan, ΔdegQ) (Fig. S1) and confirmed that $P_{degQ}$ of the comQ::kan mutant strain carries the point mutation identical to that in *B. subtilis* 168, while promoter sequences in the ΔcomQ and ΔdegQ mutants are identical to those in *B. subtilis* NCIB 3610 and *B. subtilis* PS-216 wild-type strains.

Subsequently, we verified whether $P_{srfAA}$ is responsive to comQ::kan mutation in strains carrying degQ gene under the IPTG-inducible promoter ($P_{hyperspank}$) (Fig. S2). Expression derived from $P_{srfAA}$ in the wild-type strain and the comQ::kan mutant exposed to exogenous ComX was reduced in the presence of IPTG when the deqQ gene is overexpressed. The degQ::tet mutant with the $P_{hyperspank}$-degQ construct also responded in the same way in the presence of IPTG. These results strongly support the conclusion that decreased expression of degQ due to the deletion or single-nucleotide polymorphism in its promoter (change from T to C at position −10 in the promoter region) is the main reason for the hypersensitivity of the comQ::kan strain to ComX. Although ComX is still required to activate transcription from the $P_{srfAA}$ directly, it does so with greater efficiency in strains with decreased levels of DegQ.

## Increased QS response of strain with $P_{deg}$ point mutation is not due to increased levels of QS receptor transcript

To verify whether ComP overexpression also contributes to this hypersensitivity to ComX as hypothesized previously (7), the transcriptional levels of two downstream genes: comP (encoding the ComX receptor) and comA (encoding the transcriptional activator) were quantified (Fig. S3). The results confirmed higher comP transcript abundance in the comQ::kan mutant than in the ΔcomQ mutant or the wild-type strain (Fig. S3). In contrast, the level of comA transcript was not significantly different in all tested strains (Fig. S3). To further test whether overexpression of comP in the comQ::kan strain contributes to its hypersensitivity to ComX, we introduced a $P_{hyperspank}$-comP construct into the wild-type strain (Fig. S4), which was previously confirmed to complement the comP mutant strain (7). We then measured expression derived from $P_{srfAA}$ in the presence of IPTG and found no difference between the wild-type strain and the ΔcomQ mutant supplemented with exogenous ComX, regardless of whether the strains carried $P_{hyperspank}$-comP construct or not (Fig. S4). Similarly, the comQ::kan mutant supplemented with exogenous ComX, although it still showed two- to threefold higher expression from $P_{srfAA}$ compared with the wild-type or ΔcomQ strains, did not respond to the comP overexpression. This supports the conclusion that ComX hypersensitivity of the comQ::kan strain and its higher activity of $P_{srfAA}$ is not due to increased levels of ComP.

## Hypersensitivity of signal-negative mutant *comQ::kan* manifests in increased expression of surfactin operon also during biofilm formation

In many bacteria, QS system regulates the secretion of extracellular degradative enzymes that are considered to be shareable or public goods (31). In *B. subtilis*, the ComQXPA QS system also activates transcription of the degQ gene, and DegQ positively controls extracellular degradative enzymes production via a two-component regulatory system, DegU/DegS (19, 23, 24, 26). The link between the QS system and the DegU-dependent synthesis of extracellular protease AprE (25) was investigated previously (26). Specifically, the QS signal ComX restored aprE expression in the ΔcomQ mutant but not in the degQ::tet or the ΔcomQ degQ::tet mutants (27) confirming that ComX works upstream of the DegQ protein (24). Moreover, lack of DegQ contributes to the de-repression of the srfA operon (28, 29). We here tested whether DegQ acts as an intracellular link between the ComX-sensing and the QS response, which has not been directly addressed previously. So far, all the earlier experiments have been performed in planktonic cultures using the competence-stimulating CM medium. Next, we tested whether hypersensitivity of $P_{srfAA}$ activity in the ΔcomQ::kan occurs during biofilm development. We grew all strains under static conditions in MSgg medium, which is routinely used to culture *B. subtilis* biofilms (8, 26, 32). Under these conditions, $P_{srfAA}$ activity in the comQ::kan

mutant was high even without exogenously added ComX and resembled that of the wild-type strain. However, supplementation of ComX further increased promoter activity (Fig. 2A). In contrast, the $\Delta comQ$ mutant showed low activity from $P_{srfAA}$, which was restored to the wild-type levels by ComX supplementation (Fig. 2B). Overall, these results confirmed the $P_{degQ168}$ hypersensitivity of the $comQ::kan$ mutant also during biofilm development.

## DegQ-driven hypersensitivity rewires QS response toward full investment into surfactin and lacks protease production

Since DegQ plays a key role in extracellular protease production (e.g., AprE) in floating biofilms and ComX also regulates synthesis of exoproteases (26), we next tested $P_{aprE}$ activity in the $\Delta comQ$ and the $comQ::kan$ mutants with and without ComX supplementation. $P_{aprE}$ activity in the $\Delta comQ$ mutant, which lacks the $P_{degQ168}$ mutation, was low compared to the wild-type strain (Fig. 3A). This activity was partially restored by the addition of exogenous ComX (Fig. 3B). As expected, ComX supplementation did not restore the $P_{aprE}$ activity in the $degQ::tet$ and $\Delta comQ$ $degQ::tet$ mutants (Fig. 3B). In contrast, $P_{aprE}$ activity in the $comQ::kan$ mutant only marginally responded to ComX and was similar to that in the $degQ::tet$ and $\Delta comQ$ $degQ::tet$ mutants regardless of the presence or absence of ComX (Fig. 3C and D). Intriguingly, these results suggest that the $comQ::kan$ mutant harbors differential promoter activation in response to ComX; hyperactivation of $P_{srfAA}$ (Fig. 2A and B) and decreased $P_{aprE}$ activity even in the presence of ComX. This supports the hypothesis that DegQ serves as a link between signal sensing and the adaptive response (e.g., surfactin and protease production), which assures optimal distribution of the investment toward two different public goods, surfactin and exoproteases in floating biofilms. If this link is broken, the distribution is unequal, which decreases the fitness of the $\Delta degQ$ mutant (Fig. S5).

## DISCUSSION

We have previously demonstrated that a pleiotropic constraint at the level of ComX synthesis makes the signal-deficient mutant ($comQ::kan$) hypersensitive to the signal, which leads to overproduction of surfactin, and thus increased metabolic costs, and consequently reduced fitness of the mutant when exposed to the signal produced by

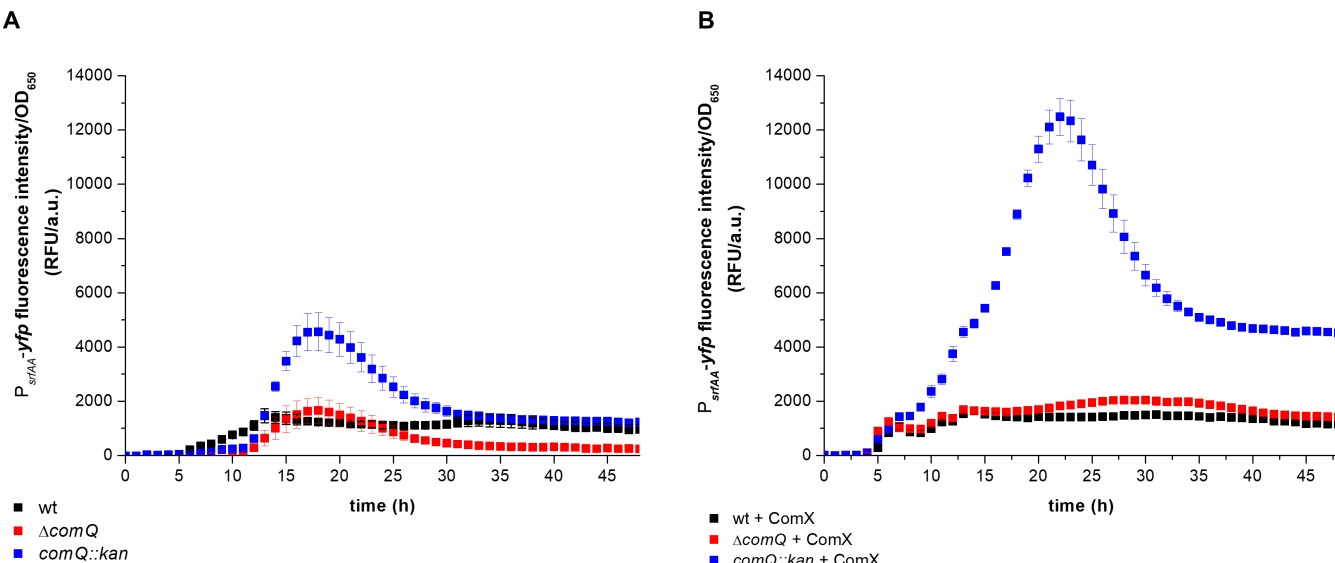

**FIG 2** Expression (fluorescence intensity normalized per optical density at 650 nm) derived from the $P_{srfAA}$-$yfp$ constructs of *Bacillus subtilis* PS-216 wild-type strain and its mutants during static incubation in MSgg medium at 37°C without ComX (A) and with ComX-conditioned 20% (vol/vol) M9 minimal medium, where ComX is produced heterologously in *E. coli* ED367 after IPTG induction (B). The values presented are means and standard errors ($n = 3$).

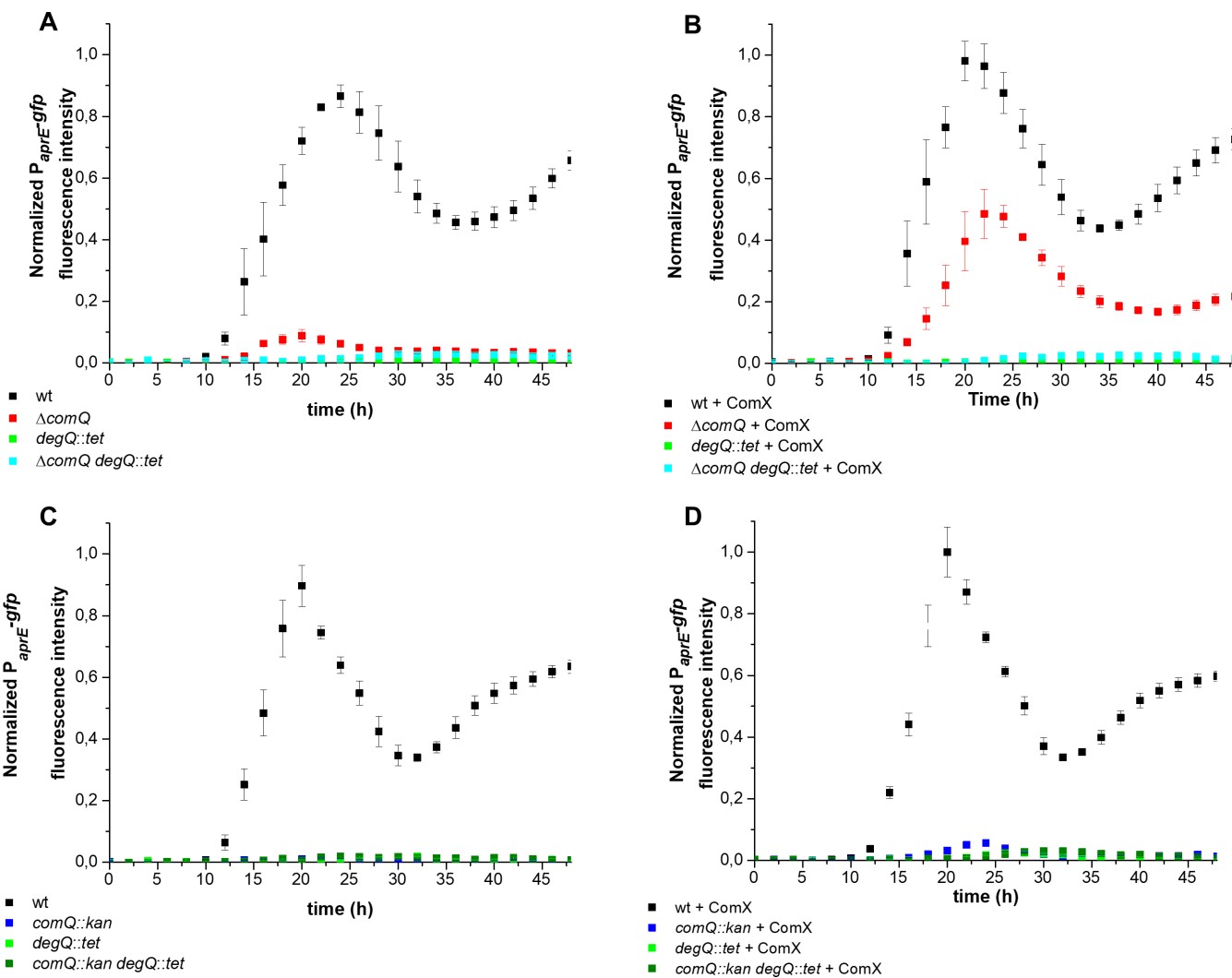

**FIG 3** Normalized expression derived from the P$_{aprE}$-*gfp* constructs of *Bacillus subtilis* PS-216 wild-type strain and its mutants during static incubation in MSgg medium at 37°C without ComX (A and C) and with ComX-conditioned 20% (vol/vol) M9 minimal medium, where ComX is heterologously produced in *E. coli* ED367 after IPTG induction (B and D). Exogenous ComX was added to the cultures 8 h after static incubation in MSgg medium. The values presented are means and standard errors (*n* = 3).

the wild-type strain produced (5). Bareia and colleagues (7) suggested that the signal-deficient mutation in the *comQ::kan* strain has a polar effect, which leads to increased expression of the downstream *comX* and *comP* genes. However, the polar effect of the Δ*comQ* mutant on the expression from P$_{srfAA}$ has not been tested (7). To further explain the hypersensitivity of the *comQ::kan* mutant to ComX, we compared its QS response to that of the signal-deficient mutant without an antibiotic resistance marker (Δ*comQ*) and, therefore, without the polar effect. Indeed, we identified a remarkable difference in the responses of these strains to ComX, with Δ*comQ* lacking hypersensitivity toward ComX. Although our results are in agreement with previous work that reported increased transcription of the *comP* in *comQ::kan* mutant (7), they also point to an important difference. Specifically, we showed that increased transcription of the receptor gene is not the main reason for the hypersensitivity and fitness disadvantage of the *comQ::kan* strain in the presence of ComX because IPTG-inducible expression of *comP* from the P$_{hyperspank}$ promoter did not render the strain hypersensitive to ComX. We, therefore, conclude that the polar effect of the kanamycin resistance cassette insertion, which leads to increased expression of *comP* gene (7), is not sufficient to explain hypersensitivity

of the *comQ::kan* mutant. Instead, we provide evidence here that the point mutation from T to C at position −10 in $P_{degQ}$ is responsible for hypersensitivity of *comQ::kan* mutant to ComX. Due to proximity of $P_{degQ}$ and *comQ* gene, the point mutation in $P_{degQ}$ was likely introduced by double homologous recombination during construction of *comQ::kan* in the PS-216 background using donor DNA from the *B. subtilis* 168 strain (16). The point mutation alone, however, was not sufficient for $P_{srfAA}$ activation during planktonic growth in CM medium. It was still dependent on ComX, which activates the ComP-ComA two-component system that is required for $P_{degQ}$ and $P_{srfAA}$ activation. The markerless Δ*comQ* strain does not harbor this mutation, and therefore the $P_{degQ}$ resembles that of the undomesticated *B. subtilis* NCIB 3610 strain (Fig. S1). When $P_{degQ168}$ was solely introduced into the PS-216 genetic background, increased surfactin production was detected in the presence of ComX (Fig. 1). This confirms that the promoter point mutation alone is sufficient for the hypersensitivity phenotype, consistent with previous findings demonstrating the importance of $P_{degQ}$ mutation for enhanced expression from $P_{srfAA}$ and transformability of the *B. subtilis* 168 strain (28). The same mutation could be potentially responsible for self-sensing in the *B. subtilis* ComQXPA system (7). This type of point mutation in $P_{degQ}$ can also easily occur spontaneously during the domestication process (33) and even in environments with high nutrients in nature (34). These genetic changes are biologically relevant as they directly indicate genome plasticity, loss of metabolically costly traits, which results in enhanced growth. Further studies addressing evolution of *degQ* under variety of environmental conditions or in natural *B. subtilis* populations will reveal the importance of this link to stabilize the signaling and the QS response.

DegQ also plays an important role in the QS regulation of extracellular degradative enzyme synthesis in a floating biofilm (30). We propose here that the *degQ* gene may represent a policing mechanism that ensures balanced production of public traits (extracellular enzymes, surfactin production) and private traits [K-state development (28)]. This hypothesis has not been tested before, and it is consistent with our results, demonstrating that exogenous ComX restores extracellular degradative enzyme synthesis in the Δ*comQ* mutant but not in the *comQ::kan* mutant, which has decreased levels of DegQ in floating biofilms. DegQ may also act as an anti-activator that prevents self-sensing in the ComQXPA QS system. Anti-activators were recently discovered for LasI/LasR QS system in *Pseudomonas aeruginosa* (35).

In conclusion, this work presents several important findings: (i) It reveals contrasting behavior between the *comQ::kan* mutant and the clean Δ*comQ* mutant. (ii) It establishes a direct link between a single-point mutation in the *degQ* promoter in the *comQ::kan* mutant and ComX-dependent overexpression of the *srfA* operon. (iii) It negates the importance of increased *comP* expression in this particular phenotype. (iv) It demonstrates that the addition of exogenous ComX fails to restore extracellular degradative enzyme synthesis in the *comQ::kan* mutant, while surfactin synthesis in this mutant is overresponsive to ComX. (v) Furthermore, it reveals that the Δ*degQ* deletion in the clean Δ*comQ* mutant also results in a similar hypersensitivity to ComX as previously shown for the *comQ::kan* mutant, even during biofilm growth. To sum up, our findings provide compelling evidence that DegQ serves as a critical intracellular link between extracellular ComX-sensing and the quorum-sensing response. By regulating extracellular degradative enzyme synthesis and surfactin production, DegQ effectively acts as a gatekeeper, preventing excessive investment in a particular public good that could prove costly to the organism. This key role of DegQ highlights its importance in maintaining a precisely balanced and efficient quorum-sensing system (Fig. 4).

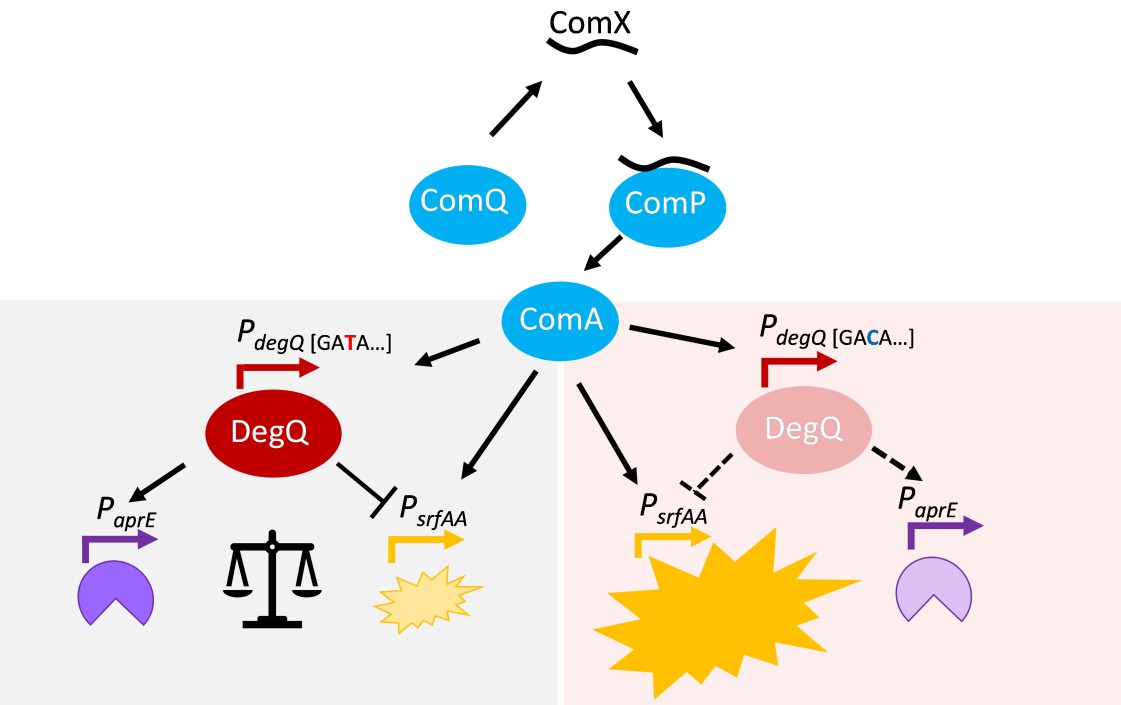

**FIG 4** Schematic presentation of the link between QS- and DegQ-regulated processes in *Bacillus subtilis*.

## MATERIALS AND METHODS

### Bacterial strains and growth conditions

Bacterial strains used in the study are listed in Table 1. Overnight cultures were incubated in liquid lysogeny broth (LB) with appropriate antibiotics at 37°C and 200 rpm. Antibiotics used were at the following final concentrations: chloramphenicol (Cm) 10 µg mL$^{-1}$, spectinomycin (Spec) 100 µg mL$^{-1}$, kanamycin (Kn) 50 µg mL$^{-1}$, tetracycline (Tet) 10 µg mL$^{-1}$, erythromycin (Ery) 0.5 µg mL$^{-1}$, lincomycin (Lin) 12.5 µg mL$^{-1}$, and ampicillin (Amp) 100 µg mL$^{-1}$.

Growth of *B. subtilis* strains in shaking conditions was measured by optical density at 650 nm (OD$_{650}$) after inoculation of fresh CM medium (40) with an overnight culture (1%, vol/vol) and incubation at 37°C and 200 rpm for 6 h.

To grow floating biofilms 1% (vol/vol) of an overnight culture was incubated in liquid MSgg medium (32) in static conditions at 37°C for up to 48 h (26). Where indicated, CM medium or MSgg medium was supplemented with 5% or 20% (vol/vol) conditioned M9 minimal medium containing ComX produced by *Escherichia coli* ED367 after IPTG induction (26). Alternatively, conditioned M9 minimal medium without ComX was prepared in the same way except for addition of IPTG to *E. coli* ED367 culture. Experiments with IPTG-inducible promoters (for P$_{hyperspank}$-*comP* and P$_{hyperspank}$-*degQ* strains) involved growth in CM medium supplemented with 0.1 mM IPTG.

### Strain construction

Mutant strains were constructed by transformation of specific markers into competent *B. subtilis* strains. Strains were grown in CM medium at 37°C and 200 rpm, and transformants were selected by antibiotic selections on LB agar plates with the appropriate antibiotics at 37°C. The Δ*comQ* and *comQ::kan* mutants were transformed by growth in CM medium supplemented with exogenous ComX (26). The *amyE*::P$_{srfAA}$-*yfp* and P$_{aprE}$-*gfp* mutants were constructed by transforming appropriate *B. subtilis* strains with genomic DNA isolated from *B. subtilis* DL722 (36) or O8G57 (37), respectively. The BM1134 mutant strain was constructed by transforming BM1400 strain

**TABLE 1** Strains used in this study

| Strain name | Descriptive | Background | Genome description | Reference |
|---|---|---|---|---|
| *Bacillus subtilis* strains | | | | |
| PS-216 | wt | | Undomesticated strain | (13) |
| BM1127 | Δ*comQ* | PS-216 | Δ*comQ* | (26) |
| BM1400 | *comQ::kan* | PS-216 | *comQ::kan* | (26) |
| BM1557 | Δ*degQ* | PS-216 | Δ*degQ* markerless | This work |
| BM1133 | *degQ::tet* | PS-216 | *degQ::tet* | (26) |
| BM1134 | *comQ::kan degQ::tet* | PS-216 | *comQ::kan degQ::tet* | This work |
| BM1445 | Δ*comQ degQ::tet* | PS-216 | Δ*comQ degQ::tet* | (26) |
| BM1454 | | PS-216 | *amyE*::P$_{srfAA}$-*yfp* (spec) | (30) |
| BM1455 | | PS-216 | Δ*comQ amyE*::P$_{srfAA}$-*yfp* (spec) | (30) |
| BM1456 | | PS-216 | *comQ::kan amyE*::P$_{srfAA}$-*yfp* (spec) | (26) |
| BM1565 | | PS-216 | Δ*degQ amyE*::P$_{srfAA}$-*yfp* (spec) | This work |
| BM1586 | | PS-216 | *sacA*::P$_{srfAA}$-*yfp* (cm) | This work |
| BM1587 | | PS-216 | Δ*comQ sacA*::P$_{srfAA}$-*yfp* (cm) | This work |
| BM1588 | | PS-216 | *comQ::kan sacA*::P$_{srfAA}$-*yfp* (cm) | This work |
| BM1589 | | PS-216 | *degQ::tet sacA*::P$_{srfAA}$-*yfp* (cm) | This work |
| BM1552 | | PS-216 | *amyE*::P$_{hyperspank}$-*comP* (spec) | This work |
| BM1553 | | PS-216 | Δ*comQ amyE*::P$_{hyperspank}$-*comP* (spec) | This work |
| BM1706 | | PS-216 | *comQ::kan amyE*::P$_{hyperspank}$-*comP* (spec) | This work |
| BM1595 | | PS-216 | *sacA*::P$_{srfAA}$-*yfp* (cm) *amyE*::P$_{hyperspank}$-*comP* (spec) | This work |
| BM1596 | | PS-216 | Δ*comQ sacA*::P$_{srfAA}$-*yfp* (cm) *amyE*::P$_{hyperspank}$-*comP* (spec) | This work |
| BM1704 | | PS-216 | *comQ::kan sacA*::P$_{srfAA}$-*yfp* (cm) *amyE*::P$_{hyperspank}$-*comP* (spec) | This work |
| BM1569 | | PS-216 | *amyE*::P$_{hyperspank}$-*degQ* (spec) | This work |
| BM1570 | | PS-216 | *comQ::kan amyE*::P$_{hyperspank}$-*degQ* (spec) | This work |
| BM1572 | | PS-216 | *degQ::tet amyE*::P$_{hyperspank}$-*degQ* (spec) | This work |
| BM1591 | | PS-216 | *sacA*::P$_{srfAA}$-*yfp* (cm) *amyE*::P$_{hyperspank}$-*degQ* (spec) | This work |
| BM1604 | | PS-216 | *comQ::kan sacA*::P$_{srfAA}$-*yfp* (cm) *amyE*::P$_{hyperspank}$-*degQ* (spec) | This work |
| BM1592 | | PS-216 | *degQ::tet sacA*::P$_{srfAA}$-*yfp* (cm) *amyE*::P$_{hyperspank}$-*degQ* (spec) | This work |
| BM1714 | P$_{degQ}$168 | PS-216 | P$_{degQ}$168 | This work |
| BM1715 | | PS-216 | P$_{degQ}$168 *amyE*::P$_{srfAA}$-*yfp* (spec) | This work |
| BM1142 | | PS-216 | P$_{aprE}$-*gfp* (cm) | (26) |
| BM1443 | | PS-216 | Δ*comQ* P$_{aprE}$-*gfp* (cm) | (26) |
| BM1143 | | PS-216 | *comQ::kan* P$_{aprE}$-*gfp* (cm) | This work |
| BM1144 | | PS-216 | *degQ::tet* P$_{aprE}$-*gfp* (cm) | (26) |
| BM1145 | | PS-216 | *comQ::kan degQ::tet* P$_{aprE}$-*gfp* (cm) | This work |
| BM1448 | | PS-216 | Δ*comQ degQ::tet* P$_{aprE}$-*gfp* (cm) | (26) |
| DL722 | | 3610 | *amyE*::P$_{srfAA}$-*yfp* (spec) | (36) |
| O8G57 | | 168 | P$_{aprE}$-*gfp* (cm) | (37) |
| *Escherichia coli* strains | | | | |
| ED367 | | BL21(DE3) | pET22(b)—*comQ comX* from *B. subtilis* 168 Amp | (11) |
| AEC1002 | | DH12 | pDR111::P$_{hyperspank}$-*comP* (Spec Amp) | (7) |
| ECE174 | | DH5α | pSac-Cm (Cm, Amp) | (38) |

(*Continued on next page*)

**TABLE 1** Strains used in this study (*Continued*)

| Strain name | Descriptive | Background | Genome description | Reference |
|---|---|---|---|---|
| ECE358 | | DH5α | pJOE8999 (Kn) | (39) |
| ED511 | | DH5α | pDR111 *amyE*::P$_{hyperspank}$ Spec Amp | Rudner, unpublished |
| ED1931 | pED1931 | DH5α | pMinimad2—PdegQ-168 (Mls, Amp) | (28) |
| EM1066 | pEM1066 | DH5α | pJOE8999::updown-degQ +sgRNA (Kn) | This work |
| EM1075 | pEM1075 | DH5α | pSac—*sacA*::P$_{srfAA}$-*yfp* (Cm, Amp) | This work |
| EM1076 | pEM1076 | DH5α | pDR111 *amyE*::P$_{hyperspank}$-*degQ* (Spec, Amp) | This work |

(26) with genomic DNA isolated from *Bacillus subtilis* BM1133 (26). The *sacA*::P$_{srfAA}$-*yfp*, *amyE*::P$_{hyperspank}$-*comP*, and *amyE*::P$_{hyperspank}$-*degQ* mutants were constructed by transforming appropriate *B. subtilis* strains with plasmid DNA pEM1075, plasmid DNA from AEC1002 (7), or plasmid DNA pEM1076, respectively.

To prepare the Δ*degQ* markerless deletion mutant, the regions upstream and downstream of the *degQ* gene were PCR amplified from genomic DNA of *B. subtilis* PS-216 using the primer pairs U-degQ-F/U-degQ-R (Table 2) or D-degQ-F/D-degQ-R, respectively (Table 2). Both fragments were digested with SfiI and EcoRI and then simultaneously ligated into the dephosphorylated SfiI site of pJOE8999 (39). The plasmid pJOE8999 carrying upstream and downstream *degQ* fragment was then digested with BsaI, and sgRNA-degQ-F/sgRNA-degQ-R (Table 2) were ligated into it to obtain pEM1066. SgRNA was designed using Benchling (Benchling Inc., USA) (41). The constructed plasmid pEM1066 was transformed into *B. subtilis* PS-216, and transformants were selected on LB agar plates containing 30 µg mL$^{-1}$ kanamycin and 0.2% (wt/vol) mannose at 30°C after incubation for 2 days. Transformants were then inoculated onto LB agar plates and incubated at 50°C overnight and then reinoculated onto LB agar plates and incubated overnight at 42°C. Colonies were then plated on LB and LB Kn (30 µg mL$^{-1}$) agar plates and incubated at 37°C to test for plasmid loss (39). Chromosomal DNA from colonies that excised the plasmid was isolated and screened by PCR using primer pairs U-degQ-F/D-degQ-R (Table 2) to determine which isolates carried a deletion in the *degQ* gene.

Bacterial strains carrying the P$_{degQ168}$ were constructed by transforming appropriate *B. subtilis* strains with plasmid DNA pED1931 (28). Plasmid-free strains were prepared according to an established protocol (44). P$_{degQ168}$ was then PCR amplified from the chromosome using primer pair PdegQ-F/PdegQ-R (Table 2) and sequenced by Macrogen Europe B. V., Netherlands to confirm the point mutation.

To prepare pEM1075 plasmid carrying *sacA*::P$_{srfAA}$-*yfp*, the P$_{srfAA}$-*yfp* region was PCR amplified from genomic DNA of *B. subtilis* DL722 (36) using the primer pair PsrfAA-EcoRI/oDR78 (Table 2). The fragment was then digested with EcoRI and BamHI and ligated into EcoRI and BamHI sites of pSac-Cm (38). To construct pEM1076 plasmid, carrying *amyE*::P$_{hyperspank}$-*degQ*, the *degQ* gene was PCR amplified from genomic DNA of *B. subtilis* PS-216 using the primer pair degQ-F-SalI/degQ-R-PaeI (Table 2), and the fragment was digested with SalI and PaeI and ligated into SalI and PaeI sites of pDR111. Both constructed plasmids were then transformed into competent *E. coli* DH5α cells, and transformants were selected on LB agar plates containing 100 µg mL$^{-1}$ ampicillin after overnight incubation at 37°C. Plasmids were isolated and screened by PCR using the primer pairs listed earlier to determine which plasmids carried the *amyE*::P$_{hyperspank}$-*degQ* or *sacA*::P$_{srfAA}$-*yfp* construct before transformation in *B. subtilis* strains.

## Expression derived from the P$_{srfAA}$-*yfp* construct in different *B. subtilis* strains in CM medium

Fresh CM medium was inoculated with overnight cultures (1%, vol/vol) and incubated at 37°C and 200 rpm for 6 h. CM medium was sometimes supplemented with exogenous ComX-conditioned (5%, vol/vol) M9 minimal medium containing ComX heterologously produced in *E. coli* ED367 after IPTG induction (26). After incubation, 200 µL aliquots were dispensed into sterile 96-well black transparent-bottom microtiter plates in three

**TABLE 2** Oligonucleotides used in this study

| Oligonucleotide name | Sequence 5′–3′ | Reference |
|---|---|---|
| U-degQ-F | GGCCAACGAGGCCCACCTGCTACATTTGCTAGTGC | This work |
| U-degQ-R | CGGAATTCACGACAGATTCATTACGAAACATT | This work |
| D-degQ-F | CGGAATTCTTCCATCGTTTCCACACTCC | This work |
| D-degQ-R | GGCCTTATTGGCCCGGCTTTGCGTTCCGATAAG | This work |
| sgRNA-degQ-F | TACGCTTTAATATCAAGTTCGAGT | This work |
| sgRNA-degQ-R | AAACACTCGAACTTGATATTAAAG | This work |
| degQ-F-SalI | ACGCGTCGACCGGTGAAAAATGAGCCGAAAGC | This work |
| degQ-R-PaeI | ACATGCATGCCTGCTCAATAACGACTTCCCCC | This work |
| PdegQ-F | CGTTTCCACACTCCTTTTTTTGAA | This work |
| PdegQ-R | TAGATCCCTAATTGCCGAATATG | This work |
| PsrfAA-EcoRI | CGGAATTCGCTATATGGAATTGATTGATATCG | This work |
| oDR78 | GCCGGATCCTTATTTGTATAGTTCATCCATGCC | (42) |
| oTB42 | AGGATTGGAAGCTGTTCGT | (43) |
| oTB43 | TGACTTCTACCGCAGGAC | (43) |
| oTB100 | AATCCGTGCCGTGAAGAG | This work |
| oTB101 | GAACCCGCTCCTGCTG | This work |
| oTB102 | GAGTTTCTGTCTAATGCGGT | This work |
| oTB103 | CGCCCATCTAAAGCCCT | This work |
| oTB104 | TTCGAGGAAGCGATTCGTG | This work |
| oTB105 | TTTGAGAGGAAGGAGCCG | This work |

technical replicates to immediately measure $OD_{650}$ and fluorescence intensity in a Cytation 3 imaging reader (BioTek, USA). Fluorescence intensity of yellow fluorescent protein (YFP) was used to monitor expression derived from the $P_{srfAA}$-$yfp$ construct with excitation at 500 nm, emission at 530 nm, and the gain set to 100. As a control, the same strains without the fluorescent marker were cultured. Fluorescence intensity was normalized per $OD_{650}$ of the strain. To calculate final expression, normalized autofluorescence of unmarked strains was subtracted from that of the marked strains.

## Expression derived from the $P_{srfAA}$-$yfp$ and $P_{aprE}$-$gfp$ constructs in different *B. subtilis* strains during growth in MSgg medium in static conditions

Briefly, fresh MSgg medium was inoculated with overnight cultures (1%, vol/vol) of fluorescently marked and unmarked strains, and 200 µL aliquots were dispensed in a sterile 96-well black transparent-bottom microtiter plate in four technical replicates. MSgg medium was supplemented with exogenous ComX-conditioned (20%, vol/vol) M9 minimal medium containing ComX heterologously produced in *E. coli* ED367 after IPTG induction (26) at the beginning of incubation, where indicated. The lid of microtiter plate was sealed with micropore tape, and the space between wells was filled with sterile deionized water to minimize the effect of medium evaporation. The microtiter plate was incubated in the Cytation 3 imaging reader (BioTek, USA) at 37°C without shaking for 48 h. Optical density at 650 nm and fluorescence intensity were measured in 30 min intervals during incubation. Fluorescence intensity of YFP was used to monitor expression derived from the $P_{srfAA}$-$yfp$ with excitation at 500 nm, emission at 530 nm, and the gain set to 100. Fluorescence intensity of GFP (green fluorescent protein) was used to monitor expression derived from the $P_{aprE}$-$gfp$ with excitation at 480 nm, emission at 510 nm, and the gain set to 50. To calculate the final expression, the fluorescence intensity of unmarked strains was subtracted from the fluorescence intensity of the marked strains. Fluorescence intensity was then normalized per $OD_{650}$ of fluorescently marked strains at each time point. Expression derived from the $P_{aprE}$-$gfp$ constructs of all tested strains was further normalized on the maximal normalized value calculated for the wild-type strain supplemented with exogenous ComX in each setup of experiments.

## $P_{degQ}$ sequence determination in different *B. subtilis* strains

$P_{degQ}$ nucleotide sequence was determined by isolating chromosomal DNA from different *B. subtilis* PS-216 strains (wt, Δ*comQ*, *comQ::kan,* and Δ*degQ*) and PCR-amplifying the fragment of interest using the primer pair PdegQ-F/PdegQ-R (Table 2). $P_{degQ}$ fragments were sequenced by Macrogen Europe B.V., Netherlands, then aligned using MUSCLE in UGENE program version 1.30.0 (45, 46) and compared with sequences in the NCBI database (e.g., *B. subtilis* 168, *B. subtilis* NCIB 3610).

## Transcript levels of *comP* and *comA* genes determination

Transcript levels of *comP* and *comA* genes in *B. subtilis* PS-216 wt, Δ*comQ,* and *comQ::kan* were determined in stationary phase bacterial cells grown in CM medium at 37°C and 200 rpm. Total RNA was isolated as previously described (47) using four independent biological replicates for each strain. Briefly, cell pellets were frozen in liquid nitrogen and stored at −80°C. Subsequently, RNA was extracted using the Macaloid/Roche protocol, and purified RNA samples were treated with RNase-free DNase I (Thermo Fisher Scientific, Germany) for 60 min at 37°C in Dnase I buffer [10 mM Tris·HCl (pH 7.5), 2.5 mM $MgCl_2$, 0.1 mM $CaCl_2$]. Samples were re-purified with the Roche RNA Isolation Kit. Reverse transcription was performed with 50 pmol random nonamers on 2 µg of total RNA using RevertAidTM H Minus M-MuLV Reverse Transcriptase (Thermo Fisher Scientific, Germany). Quantification of cDNA was performed on a MiniOpticon real-time PCR (BioRad, Hercules, CA) using Maxima SYBR Green qPCR Master Mix (Thermo Fisher Scientific, Germany). The amount of *comP1*, *comP2,* and *comA1* cDNA was determined using primer pairs oTB100/oTB101, oTB102/oTB103, and oTB104/oTB105, respectively (Table 2). The amount of target cDNA was normalized to the level of reference gene *girB* cDNA using primer pair oTB42/oTB43 (43) (Table 2) as described previously (48).

## Statistical analysis

Data are presented as means and standard errors of at least three biological replicates. Results were statistically evaluated using one-way ANOVA followed by Bonferroni's post-hoc comparison tests using the $P \leq 0.05$ significance level.

## ACKNOWLEDGMENTS

We thank Ramses Gallegos-Monterossa for his help with molecular techniques, David Dubnau for providing ED strains, and Avigdor Eldar for AEC strain.

The work has been funded by Slovenian Research Agency ARRS program grant No. P4-0116 and project grants J4-8228, J4-9302.

All authors have declared that there is no conflict of interest.

## AUTHOR AFFILIATIONS

[1]Department of Microbiology, Chair of microbial ecology and physiology, University of Ljubljana, Biotechnical Faculty, Ljubljana, Slovenia
[2]Department of Biotechnology and Biomedicine, Bacterial Interactions and Evolution Group, Technical University of Denmark, Kongens Lyngby, Denmark

## AUTHOR ORCIDs

Tjaša Danevčič http://orcid.org/0000-0002-4076-9901

## FUNDING

| Funder | Grant(s) | Author(s) |
|---|---|---|
| Javna Agencija za Raziskovalno Dejavnost RS (ARRS) | P4-0116 | Tjaša Danevčič |
| | | Mihael Spacapan |

| Funder | Grant(s) | Author(s) |
|---|---|---|
| | | Anna Dragoš |
| | | Ines Mandic-Mulec |
| Javna Agencija za Raziskovalno Dejavnost RS (ARRS) | J4-8228 | Tjaša Danevčič |
| | | Mihael Spacapan |
| | | Ines Mandic-Mulec |
| Javna Agencija za Raziskovalno Dejavnost RS (ARRS) | J4-9302 | Tjaša Danevčič |
| | | Mihael Spacapan |
| | | Ines Mandic-Mulec |

## AUTHOR CONTRIBUTIONS

Mihael Spacapan, Conceptualization, Data curation, Methodology, Visualization, Writing – original draft | Anna Dragoš, Data curation, Methodology, Writing – review and editing | Ákos T. Kovács, Data curation, Methodology, Visualization, Writing – review and editing | Ines Mandic-Mulec, Conceptualization, Funding acquisition, Investigation, Writing – original draft.

## ADDITIONAL FILES

The following material is available online.

### Supplemental Material

**Figures S1 to S5 (Spectrum00908-23-s0001.docx).** Figures S1 to S5

### Open Peer Review

**PEER REVIEW HISTORY (review-history.pdf).** An accounting of the reviewer comments and feedback.

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
