## [Reviewer comments · Microbiology Spectrum]

Microbiology Spectrum

DegQ is an important policing link between quorum sensing and regulated adaptative traits in *Bacillus subtilis*

Tjasa Danevcic, Mihael Spacapan, Anna Dragoš, Ákos T Kovács, and Ines Mandic-Mulec

Corresponding Author(s): Ines Mandic-Mulec, University of Ljubljana, Biotechnical Faculty

Review Timeline:

Submission Date:	March 1, 2023
Editorial Decision:	March 17, 2023
Revision Received:	May 30, 2023
Editorial Decision:	June 22, 2023
Revision Received:	July 25, 2023
Accepted:	July 26, 2023

Editor: Ilana Kolodkin-Gal

Reviewer(s): The reviewers have opted to remain anonymous.

Transaction Report:

DOI: <https://doi.org/10.1128/spectrum.00908-23>

Prof. Ines Mandic-Mulec
University of Ljubljana, Biotechnical Faculty
Department of Microbiology
Večna pot 111
Ljubljana 1000
Slovenia

Re: Spectrum00908-23 (DegQ is an important policing link between quorum sensing regulated adaptative traits in *Bacillus subtilis*)

Dear Prof. Ines Mandic-Mulec:

I have received the reviews of your manuscript entitled "DegQ is an important policing link between quorum sensing regulated adaptative traits in *Bacillus subtilis*", and I regret to inform you that we will not be able to publish it in Spectrum. Your submission was read by reviewers with expertise in the area addressed in your study and it was the consensus view of these reviewers that your paper did not meet the standards necessary for publication. Copies of the reviewers' comments are attached for your consideration.

Unfortunately, feedback from the reviewers and particularly reviewer 2 is discouraging. The common main concern of both reviewers addressed the lack of novelty and that the role of DegQ in the *srfAA* activities has been investigated in previous studies.

I am sorry to convey a negative decision on this occasion, but I hope that the enclosed reviews are useful. Please note, rejections from Microbiology Spectrum are final and your manuscript will not be considered by other ASM journals. We wish you well in publishing this report in another journal and hope that you will consider Spectrum in the future.

Sincerely,

Ilana Kolodkin-Gal
Editor, Microbiology Spectrum

Reviewer comments:

Reviewer #1 (Comments for the Author):

In this study, Danevcic et al investigated the role of DegQ in linking ComQXPA-mediated quorum-sensing (QS) and QS/DegQ-controlled surfactin production, production of extracellular enzymes, and genetic competence. The authors also successfully clarified some previously unsolved issues about the difference between the two different *comQ* mutants in response to extracellularly added QS signal ComX. The authors' group has made some very interesting discovery on the role of ComQXPA-mediated QS in cell-cell communication in *B. subtilis*. This is another interesting study on the same topic, focusing on the role of QS-controlled *degQ* gene. The results are well presented. The writing of the manuscript is fine, but clarity in some places in the manuscript could be improved.

I do have two issues about this study.

First, the authors have revisited the role of *degQ* in the activities of *srfAA* genes, and surfactin production, and production of extracellular protease AprE. Many of these regulations have been reported in published studies (the authors should cite those studies well). Therefore, the significance of this work, in my view, is mostly about connecting the dots and presenting those regulations in new contexts. There are some interesting ideas about evaluating the role of DegQ in fitness. For what was revealed in this study, mechanistic novelty could be something one can argue about.

The second issue is that the authors are trying to highlight the biological significance of some observations while the root reason of the described phenomenon is technical.

For example, the *comQ::kan* insertional mutant is apparently a less suitable strain for this study for technical reasons, but the authors have been comparing the *comQ::kan* mutant to the *comQ* in-frame deletion strain throughout this study. While I agree that when the authors initially tried to characterize the difference between the two mutants, both strains are needed, but after the initial characterization, the authors should have probably proceeded with the in-frame deletion mutant only for the rest of this work.

As another example, the point mutation in the degQ promoter is a recently acquired mutation due to laboratory manipulations (and it becomes relevant in this study due to transformation), thus discussing biological significance of such a mutation does not make too much sense, unless the degQ promoter polymorphism is a widely present phenomenon in *Bacillus subtilis*.

Overall, the results in this study are largely solid and presented well, but the biological significance should be downplayed in various sections in this manuscript due to the above discussed reasons.

Minor points:

Line 48, "This adaptive response helps bacteria to collectively cope with high population density". QS is induced at high population density, but not to cope with high population density as its goal.

Line 50, but recently they are also being applied as models to...

Line 58, consequently induction of the ComA regulon.

Line 60, which ultimately prevents ComK degradation and...

Lines 63-65, "Our previous work demonstrated that the QS signal-deficient strain of *B. subtilis* (comQ::kan) has a decreased surfactin production and despite low metabolic cost of ComX production, it grows faster than the wild-type strain." This argument seems flawed; it grows faster probably because of reduced levels of surfactin production and production of large surfactin biosynthetic enzymes and maybe exoenzymes, etc. These are high metabolic cost activities. I doubt how many people will think it has to do with producing ComX.

Line 68, "However, the mechanisms leading to this unusual phenotype of signal deficient is not completely understood" I don't quite understand this statement (not until I read further into the paragraph), what is "this unusual phenotype of signal deficient"?

Line 70, Chemical complementation (by ComX) of the comQ mutant is expected to decrease the fitness of the mutant to a level comparable to the wild type (see my point in lines 63-65). So this will not be something surprising or unexpected unless the decreased fitness is significantly lower than even the wild type, and the fitness of the mutant by chemical complementation also depends on how much ComX is added, possibly dose-dependent.

Line 88, "that the degQ locus, while unlikely the increased comP transcription, leads to"

The word "hypersensitivity", chemical complementation may have a dose issue, the authors may need to be cautious that the hypersensitive response (compared to the wild type) is not due to higher doses of ComX in the chemical complementation.

The role of DegQ in the srfAA activities has been investigated in previous studies. It is likely mediated by DegU, whose activities are modulated by DegQ. I guess it is not entirely clear yet, but the authors need to add those references and even in the introduction section.

Line 240, "To sum up, we demonstrated that by regulating extracellular degradative enzyme synthesis and surfactin production, DegQ acts as the private link between ComX production and the response (Figure 6)."

This is one example about the main issue I raised above of this study, even though the model itself (Figure 6) is interesting. What have already been known in published studies include: DegQ regulates exoproteases (AprE), srfAA operon and surfactin production; degQ is regulated by ComQXPA, so technically none of these was really discovered in this study.

Reviewer #2 (Comments for the Author):

Unfortunately, this manuscript was written very poorly. I did struggle with it and compared it to what is already known. Basically, the authors have shown that a mutation in the lab strain of *B. subtilis* has decreased the expression of degQ and that this has a positive effect on the expression of srfA. But this was already known. They do clarify the phenotypic difference between a comQ clean deletion and a comQ::kan knockout, which seems to be the only novel finding in this report. But this does not justify a publication. I am not sure what is meant by a "private link" and how this term adds anything. I found it confusing. I am sorry to be so negative and it may be that I have missed something important, due to the confusing presentation.

June 22, 2023

Prof. Ines Mandic-Mulec
University of Ljubljana, Biotechnical Faculty
Department of Microbiology
Večna pot 111
Ljubljana 1000
Slovenia

Re: Spectrum00908-23R1-A (DegQ is an important policing link between quorum sensing and regulated adaptative traits in *Bacillus subtilis*)

Dear Prof. Ines Mandic-Mulec:

As novelty is not a requirement for spectrum, and the previous major concerns were addressed, the authors can now proceed with the final revision. However, I'd like to ask the authors to extend the discussion on the importance of this particular study in light of previous findings.

Link Not Available

Sincerely,

Ilana Kolodkin-Gal

Journals Department
Reviewer comments:

Reviewer #1 (Comments for the Author):

The major finding in this study presented by Danevcic et al is to clarify some of their previous observations about the differential responses of two different comQ mutant strains to the ComQXPA QS. Through this study, the authors concluded that the difference is due to the point mutation in the degQ promoter from the domesticated strain 168 that was unpurposely introduced into the strains applied in this study. This is a resubmitted manuscript in which the authors make overall changes especially in

the result section to address previous review comments. The authors clarified the role of DegQ in some of the ComQPA controlled activities, which should help us better understand the discrepancy long observed among different lab strains used in the field.

The authors also proposed "degQ as the private link". I am not sure if I agree with the concept of the private link here. It has been shown that DegQ regulates the activity of the DegUS two component system, which happens to regulate the srfA operon that is controlled by ComAP as well. There is a link between QS and DegQ, but again it is not a novel function of DegQ beyond regulating DegUS, and why "private"? maybe takes out or rephrase the word "private" ?

Minor points:

Line 29, ...which is under the positive control by DegQ.

Line 54, the authors can spell out the full name, *Bacillus subtilis*, here since it is the first time showing up in the introduction section, whereas no need in line 78.

Line 79, the ComQPA QS system...

Line85, ...by increasing the transcription levels of the comP gene.

Line147, the level of comA transcript...

Staff Comments:

Preparing Revision Guidelines

Please return the manuscript within 60 days; if you cannot complete the modification within this time period, please contact me. If you do not wish to modify the manuscript and prefer to submit it to another journal, please notify me of your decision immediately so that the manuscript may be formally withdrawn from consideration by Microbiology Spectrum.

Response to Reviewers

Dear Prof. Ines Mandic-Mulec:

As novelty is not a requirement for spectrum, and the previous major concerns were addressed, the authors can now proceed with the final revision. However, I'd like to ask the authors to extend the discussion on the importance of this particular study in light of previous findings.

We have added a few additional sentences in the discussion section of the revised manuscript and slightly modified the text to smooth it out (lines 243-256).

Link Not Available

Sincerely,

Ilana Kolodkin-Gal

Journals Department
Reviewer comments:

Reviewer #1 (Comments for the Author):

The major finding in this study presented by Danevcic et al is to clarify some of their previous observations about the differential responses of two different comQ mutant strains to the ComQXPA QS. Through this study, the authors concluded that the difference is due to the point mutation in the degQ promoter from the domesticated strain 168 that was unpurposely introduced into the strains applied in this study. This is a resubmitted manuscript in which the authors make overall changes especially in the result section to address previous review comments. The authors clarified the role of DegQ in some of the ComQPXA controlled activities, which should help us better understand the discrepancy long observed among different lab strains used in the field.

The authors also proposed "degQ as the private link". I am not sure if I agree with the concept of the private link here. It has been shown that DegQ regulates the activity of the DegUS two component system, which happens to regulate the srfA operon that is controlled by ComAP as well. There is a link between QS and DegQ, but again it is not a novel function of DegQ beyond regulating DegUS, and why "private"? maybe takes out or rephrase the word "private" ?

We have removed the word private in the revised manuscript.

Minor points:

Line 29, ...which is under the positive control by DegQ.

We made correction in the revised manuscript as suggested.

Line 54, the authors can spell out the full name, Bacillus subtilis, here since it is the first time showing up in the introduction section, whereas no need in line 78.

We made correction in the revised manuscript as suggested.

Line 79, the ComQXPA QS system...

We made correction in the revised manuscript as suggested.

Line85, ...by increasing the transcription levels of the comP gene.

We made correction in the revised manuscript as suggested.

Line147, the level of comA transcript...

We made correction in the revised manuscript as suggested.

July 26, 2023

Prof. Ines Mandic-Mulec
University of Ljubljana, Biotechnical Faculty
Department of Microbiology
Večna pot 111
Ljubljana 1000
Slovenia

Re: Spectrum00908-23R2 (DegQ is an important policing link between quorum sensing and regulated adaptative traits in *Bacillus subtilis*)

Dear Prof. Ines Mandic-Mulec:

Your manuscript has been accepted, and I am forwarding it to the ASM Journals Department for publication. You will be notified when your proofs are ready to be viewed.

Sincerely,

Ilana Kolodkin-Gal
Editor, Microbiology Spectrum
